# A Systematic Overview of Eudragit^®^ Based Copolymer for Smart Healthcare

**DOI:** 10.3390/pharmaceutics15020587

**Published:** 2023-02-09

**Authors:** Aniket Nikam, Priya Ranjan Sahoo, Shubham Musale, Roshani R. Pagar, Ana Cláudia Paiva-Santos, Prabhanjan Shridhar Giram

**Affiliations:** 1Department of Pharmaceutical Chemistry, Dr. D. Y. Patil Institute of Pharmaceutical Sciences and Research, Pimpri, Pune 411018, India; 2Department of Chemistry, University of Delhi, Delhi 110007, India; 3Department of Chemistry, University at Buffalo, The State University of New York, Amherst, NY 14260, USA; 4Department of Pharmaceutics, Dr. D. Y. Patil Institute of Pharmaceutical Sciences and Research, Pimpri, Pune 411018, India; 5Department of Pharmaceutical Technology, Faculty of Pharmacy of the University of Coimbra, 3004-531 Coimbra, Portugal; 6REQUIMTE/LAQV, Group of Pharmaceutical Technology, Faculty of Pharmacy of the University of Coimbra, 3004-531 Coimbra, Portugal; 7Department of Pharmaceutical Sciences, University at Buffalo, The State University of New York, Buffalo, NY 14214, USA

**Keywords:** Eudragit classification, Eudragit synthesis, drug delivery, nanoparticles, biosensor

## Abstract

Eudragit, synthesized by radical polymerization, is used for enteric coating, precise temporal release, and targeting the entire gastrointestinal system. Evonik Healthcare Germany offers different grades of Eudragit. The ratio of methacrylic acid to its methacrylate-based monomers used in the polymerization reaction defines the final product’s characteristics and consequently its potential range of applications. Since 1953, these polymers have been made to use in a wide range of healthcare applications around the world. In this review, we reviewed the “known of knowns and known of unknowns” about Eudragit, from molecule to material design, its characterization, and its applications in healthcare.

## 1. Introduction

In the last decades, natural or synthetic polymers have made a significant difference in painting, cosmetics, bio-imaging, trade, and medicine. The contributions of polymer science and engineering to healthcare applications are showing promise. These polymers have many advantages in drug delivery applications, such as being biodegradable, biocompatible, simple to eliminate from the body, minimizing the number of doses needed, maintaining the drug concentration level in the optimal range, and making patients more likely to take their medicine because the polymer coat hides the organoleptic characteristics of the formulation. According to literature studies, each polymer has some unique properties that permit the fabrication of nanotechnology-based formulation with a controlled size distribution, solubility, flexibility, and permeation for the intended application.

These properties are crucial in the development of novel polymeric carriers for the preparation of nanocarrier coating test masking site-specific release time-dependent release accomplished using multiple types of polymers to alter the release pattern or modify the release kinetics in therapeutic drug delivery application. To deliver the drug at target sites, the drug was enclosed in a polymeric shell matrix. However, some polymers have a few limitations, such as being non-compatible with active ingredients, non-biodegradable, and toxic; to overcome these problems, there is a growing concern for the synthesis of polymer by modern polymerization method with catalysis. In recent years, Eudragit has proven to be one of the most attractive areas of research due to the importance of its advanced drug delivery system. Rohm & Hass GmbH, Darmstadt, first introduced Eudragit in 1953 as an acid-resistant, alkaline-soluble drug coating functional material. Eudragit is a brand name marketed primarily by Evonik Technologies Germany.

It is an extensive collection of anionic, cationic, or neutral copolymers derived from methacrylic acid and methacrylic or acrylic esters, as well as their derivatives. According to their physicochemical properties, such copolymers synthesized using diverse polymerization techniques such as reversible addition–fragmentation chain transfer (RAFT), atom transfer radical polymerization (ATRP), and group transfer exhibit varying degrees of pH-dependent or independent solubility in numerous studies. Different varieties of Eudragit polymer are utilized to selectively target the entire GIT, including the small intestine, colon, and large intestine. These copolymers are also investigated for protein and drug protection from the stomach’s acidic environment, etc. Eudragit are nonionic and synthetic polyionic copolymers, including different concentrations of methacrylic acid esters, alkyl methacrylates, 2-(dimethylamino)ethyl methacrylate. The various concentration of non-ionized and ionized groups in the structure of these copolymers generally controls. They are typically used as film-coating agents in oral tablets as well capsule formulations for protection and to give prolonged release dosage form. In both organic and aqueous wet granulation techniques, Eudragit polymers are used as binders [1].

The next paragraph discusses the nomenclature of several grades of Eudragit polymer. Eudragit E100, E12,5, and EPO are cationic copolymer made from methyl methacrylate, dimethylaminoethyl methacrylate, and butyl methacrylate in all three grades. Poly(butyl methacrylate-co-(2-dimethylamino) ethyl methacrylate-co-methyl methacrylate) is its chemical name. The alkali range is 180 mg KOH/g, the molecular weight is roughly 47,000 g/mol, and the solubility at gastric pH 5 is low viscosity. Both Eudragit S 12,5 and S 100 are anionic copolymers that are produced from methyl methacrylate and methacrylic acid. Glass transition temperature, acid value, and molecular weight is >150 °C, 190 mg KOH/g, 125,000 g/mol, and soluble in basic pH at 7.0. Soluble Eudragit FS 30D is an anionic copolymer of methacrylic acid, methyl acrylate, and methyl methacrylate. Glass transition temperature, acid value, and molecular weight is >150 °C, 190 mg KOH/g, 125,000 g/mol, and soluble in basic pH at 7.0. Soluble Eudragit FS 30D is an anionic copolymer of methacrylic acid, methyl acrylate, and methyl methacrylate. It is soluble in above pH 7 but insoluble in acid pH. Glass temperature, molecular weight, and acid value are 48 °C, 280,000 g/mol, and 70 mg KOH/g. Eudragit L12,5 and L 100 anionic derived methyl methacrylic acid and methacrylic acid have the same molecular weight, 125,000 g/mol, glass transition temperature of more than 150 °C acid value of 315 mg KOH/g, and soluble above pH 6. Eudragit RL100, RL 30 D, and RLPO are copolymers containing methyl methacrylate and ethyl acrylate with quaternary ammonium groups and less methacrylate acid. RL 30D grade has low viscosity and a milky-white appearance. Its glass transition temperatures are 40 °C and 55 °C, and its molecular weight is 32,000 g/mol. Eudragit RL 100 and RLPO share a 70 °C glass temperature, 32,000 g/mol molecular weight, and 28.1 mg KOH/g acid value [2].

Furthermore, the majority of Eudragit polymer grades are widely used for coating pharmaceutical formulations and the development of innovative drug delivery systems. In this review, we describe the Eudragit polymer, classification, chemical composition, characterization, and synthesis of Eudragit, as well as its applicability as a carrier system in multidisciplinary research on targeted drug administration. We also look at the use of functionalized Eudragit in many advanced drug delivery applications such as capsule and tablet coating, processing of microsphere nanoparticles nanofibers, and so on.

## 2. Classification of Eudragit Polymer

Many different grades of Eudragit polymer usually available are granules, dry powder, organic solvent, and aqueous dispersion forms. A mixture of acetone and isopropanol in a ratio of 40:60 is highly used as the organic solvent. The chemical compositions, characteristic feature of different types of Eudragit, are shown in Table 1.

In this table, there are four main classes of Eudragit polymers listed: cationic Eudragit E (used for taste masking and soluble below pH 5.5), anionic Eudragit L & S (used for colon targeting/enteric coating and soluble above pH 6 and 7), neutral Eudragit RL & RS (quaternary ammonium group) polymers (both of which possess pH-independent solubility), and Eudragit NE & NM are swellable and permeable for sustained release application [3].

## 3. Characterization of Eudragit

Glass transition temperature, X-ray particle diffraction, DSC, FT-IR, physiological buffer differential, and pH-sensitive properties distinguish Eudragit grades. As indicated in Table 2, a differential thermal study of these polymers indicates a single thermal start as the glass transition temperature, which is characteristic of the various Eudragit grades. The glass transition temperature affects pharmaceutical dosage from storage conditions, film production, melt processing, etc. Due to the amorphous Eudragit structure, it exhibits prolonged release. Small molecules of drugs, solvents, or plasticizers lower the glass transition temperature; these properties are important for pharmaceutical applications. Triethyl citrate is used as the main plasticizer in Eudragit [4,5]. X-ray powder diffractograms of Eudragit S 100, L 100, RS, and RL are shown in Figure 1. Based on the literature study, Eudragit grades indicate the amorphous nature of the polymers [6,7]. TGA curves with DSC thermograms of the Eudragit L30D, L as well as S are explained in Figure 2. Here, we can see thermal characteristics of the Eudragit L30D, L, and S from DSC curves appeared to be compatible with the reflectance data by DSC/FT-IR microspectroscopy. Figure 3 are presented three-dimension FT-IR spectra of Eudragit L30D, L, and S. The spectral ranges of these three polymers appear at 3100 and 2850 cm^−1^, 1800 and 1650 cm^−1^, and 1350 and 900 cm^−1^. The C-H starching bend peak range is between 3100 and 2850 cm^−1^, the C=O stretching vibration groups is between 1800 and 1650 cm^−1,^ and C-O stretching vibration mode is between 1350 and 900 cm^−1^ [8,9]. The diffraction of all these Eudragit grades shows a halo(gaussian), suggesting that the polymers are amorphous, as seen in Figure 1.

## 4. Synthesis of Eudragit Polymer

Although gastrointestinal drug release is the ideal drug delivery route, there are several problems when the medicine is injected into the digestive system. Limitation such as reduction in drug bioavailability is observed during drug transport through the gastrointestinal mucosa [10,11]. In general, drug delivery on the oral mucosa is difficult because of a constant flow of saliva and mobility of the tissue, limiting the residence time of drugs administrated to the oral cavity. The size of a buccal dosage form is restricted by the very limited area available for the application of the delivery system. This size restriction, in turn, limits the amount of drugs that can be incorporated into the dosage forms [12]. In order to improve intestinal drug transport, Haupstein et al. synthesized preactivated thiolated Eudragit L100-55 (Figure 1) [13].

The authors prepared preactivated thiolated poly(methacrylic acid-co-ethyl acrylate) (Eudragit L100-55) in two synthetic steps. In the beginning, commercially available Eudragit L100-55 was thiolated using L-cysteine through the covalent bond formation. Later, the inactivated thiol moieties were preactivated using a disulfide bond junction through the addition of 2-mercaptonicotinic acid to produce the desired target product. Various precursor compounds were formulated with different amounts of L-cysteine (particularly 60, 140, and 266 *μ*mol/g polymer) and different amounts of preactivation, such as 33, 45, and 51 µmol/g polymer. Tensile-based mucoadhesion studies resulted in 30.5, 35.3-, and 52.2-times adhesion enhancement, respectively, for preactivated Eudragit polymers. As a result, both water uptake, as well as prolonged dissolution time increased at a pH value of 6.8. The rise in mucoadhesion was attributed to the insertion of an aromatic ligand in the preactivated Eudragit polymer (Figure 2).

In an effort to transfer mucosal vaccination of protein to mucosal immune cells as part of the nasal vaccine delivery approach, Li et al. developed mannan-functionalized thiolated Eudragit microspheres in the year 2015 [14]. The commercially available Eudragit L100 was mixed with *N,N′*-dicyclohexylcarbodiimide, and *N*-hydroxysuccinimide in dimethyl sulfoxide (DMSO) solvent and stirred for one day. Further, L-cysteine hydrochloride was added to the reaction solution, which resulted in thiolated Eudragit after 2 days of stirring (Figure 3). Mannan-functionalized Eudragit microspheres increased receptor-triggered endocytosis in the antigen-presenting cells through mannose receptor stimulation.

Perez-Ibarbia et al. prepared colored polymers [15] of Eudragit L 100-55 utilizing a blue cationic dye called toluidine blue (TB) and 2-methoxy-*N*-4-phenyl-1,4-phenylenediamine (MPPD) in the presence of coupling agents such as 1-ethyl-3-(3-dimethylaminopropyl)carbodiimide (EDC) and 1,1′-carbonyldiimidazole (CDI). The authors observed well-correlated results of polymer dissolution rate with drug release. Additionally, stable solutions of dyed polymers devoid of coagulation or sedimentation were achieved after optimizing a different set of formulation studies. In vitro studies of dye-based polymers suggested no toxicity in comparison to non-modified polymers (Figure 4).

Kim et al. synthesized PEGylated Eudragit L100 polymer in two synthetic steps utilizing the most common EDC coupling in DMF. The first step involves the preparation of activated (EL-NHS ester) Eudragit L100, which subsequently reacts with mPEG-NH2 2000 or 5000 in DMF, resulting in EL-PEG 2000 and EL-PEG 5000, respectively. Further, the authors developed celecoxib-loaded proliponiosomes in order to enhance the oral delivery of the anti-inflammatory drug celecoxib. Celecoxib is essentially a non-steroidal COX-2 inhibitor. Lyophilization of celecoxib, ELP, nonionic surfactants, and phospholipid yielded amorphous solid dispersions called celecoxib-loaded proliponiosomes. The dissolution rate, as well as permeability, improved significantly in the case of celecoxib-loaded proliponiosomes in comparison to the only celecoxib suspension. Enhanced oral bioavailability was also noticed when in vivo pharmacokinetic evaluation was carried out on rat species using celecoxib-loaded proliponiosomes (Figure 5) [16].

Thymoquinone, a phytochemical compound, could be used for targeting colon cancer. Ramzy et al. prepared thymoquinone-based polymeric nanocapsules through nanoprecipitation where pH-sensitive Eudragit S100 acted as polymeric shell (Figure 6). Eudragit S100 reacted with an anisamide derivative in the presence of N,N′-Dicyclohexylcarbodiimide (DCC) to give the desired anisamide target product for sigma receptors. In vitro analysis suggested the delayed release of thymoquinone from the nanocapsules. Additionally, the polymeric nanocapsules exhibited greater cytotoxicity response in HT29 cell lines with sigma receptor overexpression.

Porfiryeva et al. developed acrylated EPO using acryloyl chloride in a single-step process. The authors confirmed the degree of acrylation using permanganatometric titration. The non-irritant properties of the synthesized acrylated polymer were evaluated using a slug mucosal irritation test. The excellent mucoadhesive response was noticed on nasal mucosa tissue in comparison to non-acrylated EPO. In another study led by Prof. Moustafine and Prof. Van den Mooter developed inter-polyelectrolyte complex utilizing two different countercharged polymers such as Eudragit EPO and Eudragit L100, within the pH range of 6–7 [17,18,19]. The in vitro swelling and drug release studies indicated that the polyelectrolyte complexes could be used for oral drug delivery. The same group also investigated inter-polyelectrolyte complex where a combination of Eudragit EPO and Eudragit S100 copolymer were tested for delivering indomethacin drug candidates. These particulate systems were able to safeguard the drug candidate from an acidic environment in the stomach [20].

Three different types of polymerization routes are discussed at the bottom.

### 4.1. Atom Transfer Radical Polymerization

Atom transfer radical polymerization (ATRP) is normally known as a transition-metal-assisted atom transfer reaction. ATRP deals with a monomer, an initiator (composed of a movable halogen group), and a catalyst (consists of a transition metal-based ligand); hence, it is considered a multicomponent system [21]. Organic derivatives such as acrylates, acrylamides, styrenes, acrylonitrile, etc., are suitable radical stabilizers during polymer synthesis and are used as monomers in ATRP reactions.

In the ATRP process, the active moieties (radicals) are produced through a transition-metal-catalyzed reversible redox reaction (Figure 7). The process involves one electron oxidation as well as the detachment of halogen species from the inactive reactant. The combination of intermediate radicals with monomers results in the propagation of the polymer chain through kp (rate constant for propagation).

### 4.2. Reversible Addition–Fragmentation

Reversible addition–fragmentation chain transfer (RAFT) polymerization is the most diverse tool to develop various functional block copolymers as it can tolerate functional monomer diversity as well as a wide variety of reaction media [22]. The synthetic process involved in RAFT polymerization is comparatively small and easy to use. Thiocarbonylthio group is a well-known chain transfer (RAFT) agent, which accelerates degenerative chain transfer in the process. Most of the chains in the RAFT polymerization contain a terminal thiocarbonylthio group, as in the case of polymer b. In other words, a monomer unit is installed between the SR bond of the chain transfer agent to produce the polymer. The mechanism of the RAPT polymerization process is shown in Figure 8.

### 4.3. Chain Transfer Polymerization

Interestingly, various polymer chains per catalyst propagate during chain transfer polymerization from catalyst1 to the chain transfer agent (Figure 9) [23]. The chain transfer occurs fast in terms of propagation and is also reversible in nature. Additionally, no other termination pathways, such as βH abstraction, take place during the polymerization process. Chain transfer metal present in the macromolecular chain facilitates chemical functionalization.

## 5. Functionalized Eudragit-Based Nanomedicine for Targeted Drug Delivery

### 5.1. Eudragit-Based Hydrogel Drug Delivery

Pharmaceutical delivery of Eudragit polymers works effectively. Eudragit hydrogels with other polymers provide prolonged drug loading, controlled release, rapid response, and plastic deformation. Eudragit-containing hydrogels adhere well to the skin without irritation. They have been used for ophthalmic, vaginal, and cutaneous delivery. Mucoadhesive formulations benefit from Eudragit, a positively charged polymer. Hydrogels improve mucosal drug adherence. Advantages over conventional methods are controlled drug release, photodegradability, and local drug delivery [24]. Dos P et al. formulated hydrogels by distributing 2% of hydroxyethyl cellulose in nanocapsule suspensions and manually mixing it with hydrogels with RS100, poly(ε-caprolactone), and Eudragit S100 nanocapsules (HG-NC-RS), for 48 h, the compositions were hydrogelled at 4 °C (Figure 4). After mixing gel and water (1:10), the pH was determined. Hydrogel surface morphology was examined with scanning electron microscopy (operating at a voltage of 10 kV).

Because of its unique dissolving behavior above pH 7, Eudragit S100 was recommended as a pH-sensitive polymer. Eudragit use in the workplace as a drug delivery vehicle has been established [26]. When Eudragit is combined with other polymers such as talc and hydroxypropylmethylcellulose, it shows stable drug loading and controlled release of drugs. In order to achieve this, Eudragit and acrylic acid were utilized to make a pH-sensitive hydrogel. When Eudragit and a cross-linking agent were added in larger amounts, swelling, drug loading, and drug release were all reduced at both acidic (pH 1.2) and basic (pH 6.8 and 7.4) pH levels, but acrylic acid had the opposite effect. Since the colon is a difficult place to aim for, authors tried to make Eudragit-co-AA pH-sensitive composite material so that losartan potassium could be provided in a manageable manner. Brief spectroscopic, structural, thermal, and morphological studies were conducted on the polymeric composites that were made. Due to these positive results and qualities, it seems likely that eudragit-co-AA hydrogel will be an effective way to deliver a wide range of hydrophilic medicines to the colon [27].

In another study, for the pH-sensitive release of tenofovir, polyelectrolyte multilayer smart vaginal films of chitosan derivatives (chitosan lactate, chitosan tartrate, and chitosan citrate) and Eudragit S100 were designed. Scanning electron microscopy and surface texture of films confirmed by SEM. The films showed a significant amount of mucoadhesion in the bovine vaginal mucosa of cows. The most promising multilayer films were made of chitosan citrate and Eudragit S100. They had no toxicity, excellent mechanical properties, moderate swelling (100%), and high mucoadhesion capacities, and the release times of tenofovir in vaginal fluid and the simulated fluid mixture were 120 and 4 h, respectively. These films are performed efficiently for patients by combination with a pH-sensitive soluble polymer (Eudragit S100) to provide formulations with pH-dependent tenofovir release, high mucoadhesion, and a mild swelling profile. With very moderate swelling and a high mucoadhesion capacity, layer-by-layer films based on a 1:1 ratio of chitosan citrate and Eudragit S100 enabled the sustained release of tenofovir in simulated vaginal fluid for up to five days as well as the quick release of all the drugs following sexual contact [28]. Diabetes-related wound refractoriness is common. Reactive oxygen species (ROS) may exacerbate chronic inflammation and slow skin wound healing. Removing reactive oxygen species from wound dressings may help chronic wounds. Alginate and positively charged Eudragit nanoparticles were used to make reactive oxygen species-containing nanocomposite hydrogel with edaravone. Eudragit nanoparticles enhanced edaravone solubility and stability, increasing its application. Combining Eudragit nanoparticles with an alginate hydrogel enhanced protection and edaravone release. The nanocomposite hydrogel improved wound healing in a dose-dependent manner. Diabetic mice recovered better with a little edaravone-loaded nanocomposite hydrogel. The scientists utilized solvent displacement and evaporation (51.77%) to produce edaravone-loaded Eudragit nanoparticles with high encapsulation. Compared to liposomes containing edaravone, which had a 23% encapsulation effectiveness, Eudragit nanoparticles significantly enhanced entrapment efficacy, showing that eudragit hydrogel could be used as a carrier for hydrophilic medicines [29,30]

### 5.2. Eudragit-Based Microneedle Drug Delivery

When compared to oral medication absorption, the transdermal route has the potential to avoid some drawbacks. The microneedle is a method for penetrating the skin’s several layers, including the stratum corneum, the skin’s principal barrier. The microneedle technique is more advanced than traditional transdermal delivery routes due to features including decreased pain, self-administration, and better patient compliance. To improve resveratrol transdermal administration, Aung et al. developed resveratrol-loaded dissolving microneedles (DMNs) with Eudragit E100, Eudragit RS100, and polyvinylpyrrolidone K90 (PVP-K90) polymers. Eudragit E100/PVP-K90 DMNs and Eudragit RS100/PVP-K90 dissolving microneedles containing 5% resveratrol were developed by micro molding. Dissolving microneedle patches composed of Eudragit RS100/PVP-K90 and Eudragit E100/PVP-K90 might be a promising drug delivery platform for resveratrol-induced hypopigmentation. The goal of this study was to develop biodegradable resveratrol-loaded dissolving microneedles based on Eudragit/polyvinylpyrrolidone to increase resveratrol transdermal delivery [31]. To accomplish this, resveratrol-containing dissolving microneedle patches were created using polymer blends of Eudragit E100 or Eudragit RS100 and PVP-K90, and their form, mechanical strength, skin insertion, ex vivo dissolution, in vitro skin permeability, and stability were assessed. Compared to a gel formulation, the resveratrol-loaded dissolving microneedle patches displayed significantly enhanced transdermal medicine administration of resveratrol because it could transfer more lipophilic molecules into skin tissue while staying stable at 25 °C for 4 weeks [32,33,34,35]. The findings indicate that dissolving microneedle patches made of Eudragit RS100/PVP-K90 and Eudragit E100/PVP-K90 may be a viable delivery platform for resveratrol hypopigmentation therapy, as well as in another study, microneedles (MNs) made of Eudragit and polyvinylpyrrolidone (PVP) and to analyze their shape, mechanical strength, and in vitro skin implantation ability [36]. After loading the model drug into the pores of the porosity polymer film using aqueous gelatin porogen, it was treated with a Eudragit S100 film to prevent drug leakage as well as provide wound pH-responsive drug release. When used simultaneously, microneedles with porous and pH-responsive polymer coverings performed much better therapeutically. The researchers encapsulated the target drug in aqueous gelatin porogen before sealing the pores with a thin layer of Eudragit S100. To encapsulate the pores and prevent the encapsulated medicine from leaking out of the pores, a thin film of Eudragit S100 was applied to the porous PLGA layer, enabling wound pH-responsive drug release.

A diagram of how the pH of the wound influences the release mechanism is shown in Figure 5. Because the pH of healthy skin is a little bit acidic, a thin layer of Eudragit S100 was put on top of porous microneedles to stop drug release at normal pH and let drugs out in wound pH environments. When the pH of the interstitial fluid is above 7 (wound pH), Eudragit S100 dissolves in it, but it does not if the microenvironment is acidic (healthy skin pH). The pH-sensitive polymer Eudragit S100 was used to coat porous polymer coatings. This stopped drug leakage and released a wound medication that was sensitive to pH. By combining the benefits of porous and pH-responsive polymer coatings on microneedles, the therapeutic results were much better. This formulation was used to study pH-sensitive drug delivery for wounds with a quick response in rats, as well as in vitro and on the skin of pigs [37]. Swellable microneedles, which are made of hydrogels and have cross-link densities that can be changed, have the ability to deliver drugs at controlled rates. In this study, swellable microneedles were made using a pharmaceutical excipient that was approved by the FDA. This was a Eudragit RL100 resin coated with ethanol to control and slow the release of the drug. The microneedles in Eudragit RL100 were also made stronger by using pore-foaming agents, which improved both the drugs released and their mechanical strength. To make drug-loaded microneedles, combine Eudragit RL100 resin with a drug that absorbs alcohol. The microneedle fabrication techniques that use Eudragit RL100 as a matrix are easy to use, do not take a long time to make, and can be adjusted for mass production. This new method is meant to be used to make medications that do not dissolve and have a controlled release. Taking this into account, Eudragit can be studied further as a potential polymer, vehicle, and combination for efficient drug delivery, such as in microneedle formulation [38].

### 5.3. Eudragit-Based Nanofiber Drug Delivery

Nanofibers improve the efficacy of the drug because nano-effects provide a large surface area, a small diameter, and high porosity [39,40]. Nanofibers can be generated at a low cost and with high efficiency using electrospinning. Polymer nanofibers with diameters ranging from a few nanometers to several micrometers can be manufactured using an electrostatically driven jet of polymer solution or melt. Electrospun nanofibers have been examined for their ability to deliver a variety of drugs [41]. Furthermore, since there is no loss throughout the operation, fibers have a high encapsulation efficiency [42].

Polyurethane and Eudragit L100-55 were utilized by Aguilar et al. for the electrospinning technique for the fabrication of nanofiber composite mats (Figure 6). Paclitaxel was effectively released in a pH-dependent manner using the regulating platform of polyurethane and Eudragit L100-55 nanocomposite mats. It was characterized by FESEM to analyze the morphology of the nanofiber composites and found that the polymer ratios affected the diameter of the nanofiber. For usage as a drug-eluting stent cover, the composite mat exhibited the necessary mechanical characteristics and in vitro cell biocompatibility [43].

Electrospinning with a single nozzle was used to make zein/Eudragit nanofibers that contained several drugs in a single step. Researchers wanted to make zein nanofibers loaded with aceclofenac and Eudragit nanofibers loaded with pantoprazole so they could test different protein materials and electrospun structures for delivering two drugs at once (Figure 7). To make fine nanofibers, different amounts of zein and Eudragit, ranging from 5% to 40%, and voltages between 10 and 25 kV were used. At a voltage of 25 kV, a mix of 20% (*w*/*v*) zein and 10% (*w*/*v*) Eudragit made the best composite fibers in the important nano-size range. Therefore, composite nanofibers made of zein and Eudragit were prepared and tested to see if they could transport medicines at the same time. When pantoprazole and aceclofenac were used together, the stomach damage caused by NSAIDs was greatly reduced. In conclusion, dual drug delivery systems reduce the number of times that nonsteroidal anti-inflammatory drugs are used for a long time and may help patients take their medicine as prescribed. These Eudragit findings can be looked at in the same way that nanofibers are made.

### 5.4. Eudragit-Based Nanoparticles Drug Delivery

Nanoencapsulation of active pharmaceutical ingredients enhances the pharmacological activity, specificity, tolerability, and therapeutic index of important drugs (nanomedicines) [45,46]. Because they are unaffected by variations in ambient pH, nanoparticles having thixotropic characteristics are superior for mucosal applications [47]. When combined with polymers such as hydroxypropyl methylcellulose and talc, Eudragit stabilized drugs and allowed for their controlled release [48]. Dominguez et al. utilized Eudragit E 100 as the polymer and the emulsification-diffusion by solvent displacement approach to synthesize a triclosan nanoparticle solution. Triclosan was molecularly dispersed throughout the batches of triclosan nanoparticles. Nanoparticles displayed improved penetration than solutions and creams [49]. Khachane et al. tested Eudragit EPO nanoparticles for meloxicam therapeutic efficacy and compared it to a standard meloxicam solution. Nanoprecipitation produced meloxicam-loaded Eudragit EPO nanoparticles. Optimized nanoparticles were anti-inflammatory and less ulcerogenic [50]. Li et al. evaluated Eudragit S 100-based enteric coatings for colon-specific drug delivery of a model drug. The enteric layer was fabricated using Eudragit S 100 and folate-chitosan nanoparticle-coated 5-fluorouracil and leucovorin [51]. Sing et al. improved oral bioavailability by designing and making atazanavir nanoparticles that were coated with Eudragit RL 100. The bioavailability of nanoparticles made from Eudragit L100 is better [52]. A modified emulsion solvent diffusion method produced Eudragit S100 (ES) nanoparticles with hydrophilic or hydrophobic model compounds (Table 3). The spherical nanoparticles had a uniform surface morphology. During in vitro studies, nanoparticles retained model compounds at vaginal pH but rapidly released them at physiological pH. Nanoparticles release drugs and are absorbed by vaginal cells. Nanoparticles have low cytotoxicity in the vaginal mucosa. This study’s pH-sensitive Eudragit S-100 nanoparticles can deliver drugs locally and systemic [53]. The mechanism was studied in detail with respect to the facilitated related mucoadhesion, drug release, and cell permeation. The thiolated Eudragit L100 nanoparticulate system exhibited a high degree of swelling at intestinal pH and a fast release of insulin in vitro. Consequently, mucoadhesive NPs contacted the surface of the mucus layer and mediated the transient opening of the tight junctions between epithelial cells, thus improving the permeability of model macromolecules across biological membranes without inducing irreversible toxicity. This made it easier for model macromolecules to pass through biological membranes without causing permanent damage. Additionally, a new nanoparticle delivery system made of biocompatible polymers and absorption enhancers seems to be a good way to make protein drugs more effective when taken by mouth [54].

## 6. Gene-Based Drug Delivery

Gene delivery system is the most important application for the successful delivery of the siRNA, DNA, RNA, Plasmid, and small genes. Several target-specific delivery approaches are discussed below.

### Eudragit-Based Drug Delivery against DNA/RNA

Gregor Doerdelmann et al. showed Eudragit nanoparticles/calcium phosphate are versatile drug and gene delivery systems for hydrophobic and hydrophilic medicines (siRNA, FITC-BSA, and siRNA/THPP). Eudragit nanoparticles/calcium phosphates have a diverse drug delivery strategy for synthetic and biomolecules, adaptable to endosomal escape and excellent cellular uptake. A cationic acrylic polymer and three cationic surfactants have been used to deliver nucleic acids with relatively low cytotoxicity. According to a study utilizing Eudragit nanoparticles, HeLa cells take up both calcium phosphate/siRNA/THPP/Eudragit and calcium phosphate/FITC-BSA (Figure 8). After 3 h, the diffuse fluorescence and individual distribution of FITC-BSA and THPP in the cytosol indicate that the nanoparticles entered the endosomes [60].

In a study explained by Esposito et al., cationic surfactant enhanced microparticle shape with Eudragit E microparticles, increasing their mean diameter with increasing concentration. The in vitro toxicity of cationic microparticles on the cultured human cell line K562 demonstrated that DDAB18-based microparticles had extremely low cytotoxicity compared to untreated cell function [61]. Nanoparticles were used to deliver protein vaccines. These nanoparticles had a core made of poly(methylmethacrylate) anionic and a shell made from Eudragit L100/55. Core-shell nanoparticles are a promising way to deliver protein subunit vaccines through parenteral and mucosal routes [62]. In another study by Bhaskar et al., cationic nanoparticles were prepared from an Alkyl methacrylate copolymer (Eudragit E100) and poly(lactide-co-glycolide), PLGA) mix to deliver plasmid DNA-encoding murine interleukin-10 intramuscularly to mice to prevent autoimmune diabetes [63]. Gemma et al. developed a controlled-release oral vaccination by microencapsulating inactivated Vibrio cholera with spray-drying methacrylic copolymers Eudragit L30D-55 and FS30D. In vitro release experiments showed that fewer than 5% of bacteria were released in the acid media, 86% by Eudragit L30D-55 microparticles in the neutral medium, and less than 30% by FS30D (Figure 9) [64].

In the study by W. Nicholas Haining et al., a model Major Histocompatibility Complex (MHC) class I-restricted peptide Silver from influenza, a matrix protein was placed inside spray-dried microparticles made of dipalmitoyl phosphatidylcholine and polymethacrylate Eudragit E100 (pH-sensitive). The peptide was released from the particle when the pH dropped to the acidic level that is commonly found in the phagosome. These microparticles could be changed to take different adjuvants in addition to the silver of interest. Encapsulation of Major Histocompatibility Complex class I epitopes in microparticles made of silver and priming of CD8+ T cells to peptide vaccines against cancer cells and viruses [65]. A literature study by Bohui Xu and co-workers developed a biodegradable polymeric carrier for oral protein vaccine delivery to Peyer’s patches antigen presentation cells. As a model protein vaccine, bovine serum albumin was loaded into mannosylated chitosan nanoparticles using tripolyphosphate ionic gelation and coated with Eudragit L100 via electrostatic interaction. In the simulated gastrointestinal fluid, spherical nanoparticles were successfully prepared with an appropriate particle size of 558.2 ± 35.6 nm, high entrapment efficacy of 90.38 ± 9.12%, and excellent stability, suitable release behavior, and great resistance to enzymatic and acid destruction. A continuous ileal loop experiment using fluorescence imaging assessed the nanoparticle’s ability to target Peyer’s patches in rats. Mannosylated chitosan nanoparticles were more accurately distributed into Peyer’s patches when Eudragit L100 was digested. Oral immunization using Bovine Serum Albumin-loaded Eudragit L100-coated mannosylated chitosan nanoparticles induced strong systemic IgG and mucosal IgA responses. These results suggested that enteric-coated mannosylated chitosan nanoparticles might deliver oral protein vaccinations. In a study by Shantanu V. Lale et al., Ragweed (Ambrosia elatior) pollen grains developed protective microcapsules. An enteric polymer, Eudragit L100-55, was applied to the inner surfaces of ragweed pollens to protect the encapsulated protein from stomach degradation and allow pH-dependent release in the intestine. Eudragit L100-55 was produced without organic solvents to prevent protein molecule degradation. BSA was used to show the concept. Adjusting Eudragit L100-55 concentrations in ragweed pollens maximized bovine serum albumin loading in the matrix. This enhanced formulation preserved bovine serum albumin in a simulated stomach acid solution (Figure 10). Ragweed-Eudragit L100-55 produced little bovine serum albumin in simulated gastric juice (pH 1.2). The formulation’s leftover bovine serum albumin was undenatured following stomach fluid exposure. Release studies in mimicking intestinal fluid (pH 6.8) showed that the solid wall of ragweed pollen offered an additional controlled release mechanism during the first several hours. In conclusion, proteins were encapsulated in ragweed pollen without denaturing them using a protein-friendly solvent for Eudragit L100-55. The formulation selectively released proteins at intestinal pH. Thus, acid labile protein medications may be orally administered using ragweed pollen and Eudragit L100-55 [66].

## 7. Cancer-Based Drug Delivery

### 7.1. Eudragit-Based Drug Delivery against Colon Cancer

Drug delivery through the colon for diseases such as irritable bowel syndrome, Crohn’s disease, and ulcerative colitis is relatively new because pH-sensitive polymers dissolve more than neutral pH. Daniela and colleagues used mesalamine containing colonic drug delivery method that was developed and tested. This is accomplished by coating a tablet core with two thin layers. In the small intestine, chitosan protects the core. The caecal material’s microbiological enzymatic behavior destroys the chitosan coating in the colon, releasing medication. Eudragit L 100 prevents the chitosan-covered core from dissolving in the gastrointestinal tract [67]. Kia Dong and colleagues used the Eudragit S 100 to prepare colon-targeting drug delivery for resin microcapsule tract-based spatial statistics-ion exchange resins (TBSS-IER) by in-liquid drying. The scanning electron microscope shows the structure and morphology of TBBS-IER and tract-based spatial statistics-digital rights management (TBBS-DRM) (SEM). Figure 11 shows the in vitro release study. TBSS-considerable DRM’s colon-targeting, in-vivo study showed a strong therapeutic impact on experimental colitis mice produced by 2, 4, 6-triniteobenzenesulfonic acid (TNBS). The resin microcapsule system’s high colon-targeting could be used to create colon-targeted drugs [68].

A study by Shiao-Wen Tsa et al. explained that Hyaluronan–cisplatin conjugate nanoparticles and Eudragit S100-coated pectinate/alginate microbeads developed a novel colonic drug delivery is a commonly developed by electrospray method with aqueous solution, the pH-dependent degradation and ligand–receptor interactions can be used to specific drug delivery in the colon. In vitro results suggested that Hyaluronan–cisplatin conjugate nanoparticles were slightly more harmful than cisplatin and Hyaluronan–cisplatin conjugate nanoparticles—pectinate/alginate microbeads with Eudragit can inhibit the release of Hyaluronan–cisplatin conjugate nanoparticles in an acidic state and release Hyaluronan–cisplatin conjugate nanoparticles under simulated colonic environment. In addition, the in vivo study Hyaluronan–cisplatin conjugate nanoparticles—pectinate/alginate microbeads encapsulated with Eudragit could be decreases cisplatin-affiliated nephrotoxicity. Hyaluronan–cisplatin conjugate nanoparticles—pectinate/alginate microbeads encapsulated by Eudragit, therefore, have the ability to be developed as effective drug delivery for the colon as a drug carrier [69]. X. Shen et al. developed a promising electrospun diclofenac sodium composite with Eudragit L 100-55 nanofibers. SEM shows particles separated from nanofiber surfaces and amorphous state nanofibers on mats characterized by XRD/DSC. In vitro dissolution tests have demonstrated that all Eudragit L 100–55 have a pH-dependent drug release profile, while nanofibers have a sustained drug release profile. Thus, drug-intensive Eudragit L 100-55 may be evolved into an oral colon drug delivery system [70]. M. Biswaranjan Subudhi and co-workers developed Eudragit S100-loaded Citrus-Pectin Nanoparticles (E-CPNs) target 5-fluorouracil (5-FU) in the colon. Colorectal cancer cells overexpress galectin-3 receptors, which citrus pectin binds to. CPNs and E-CPNs were characterized by size distribution, particle size, morphology, etc. E-CPNs released almost 70% of the medicine preferentially in the colon after 24 h in vitro. Nanoparticles were 1.5 times more cytotoxic than 5-FU solution in HT-29 cancer cell Sulphorhodamine B assays. In vivo, Eudragit S100 protected nanoparticles until they reached the colon, where they were taken up and released medicine for a long time. Thus, a sophisticated technique based on receptor-mediated uptake and pH-dependent release of E-CPNs is presented here for effective and safe colorectal cancer treatment. Eudragit S100-loaded Citrus-Pectin Nanoparticles (E-CPNs) target 5-Fluorouracil (5-FU) in the colon. Colorectal cancer cells overexpress galectin-3 receptors, which citrus pectin binds to. CPNs and E-CPNs were characterized by size distribution, particle size, morphology, etc. E-CPNs released almost 70% of the medicine selectively in the colon after 24 h in vitro. Nanoparticles were 1.5 times more cytotoxic than 5-FU solution in HT-29 cancer cell Sulphorhodamine B assays. In vivo, Eudragit S100 protected nanoparticles until they reached the colon, where they were taken up and released medicine for a long time. Thus, a sophisticated technique based on receptor-mediated uptake and pH-dependent release of E-CPNs is presented here for effective and safe colorectal cancer treatment [71].

### 7.2. Eudragit-Based Drug Delivery against Oral and Buccal Cancer

Sustained release formulation helps reduce most oral medication molecule side effects. Eudragit polymer can be utilized in sustained-release tablets because it forms a matrix structure. Badir et al. developed and evaluated vancomycin coated with Eudragit nanoparticles by w/o/w double emulsion solvent evaporation utilizing Eudragit RS as a retardant. In-vitro study showed a biphasic release pattern with a 0.5-h burst followed by a slow 24-hour release. Oral sustained delivery of vancomycin nanoparticles was successful [72]. Jingaling et al. found that nanoprecipitation of genistein nanoparticles with Eudragit E 100 carriers improves oral bioavailability. Genistein nanoparticles release twice even many drugs than capsules [73]. Momoh et al. developed Eudragit RS and RL 100 microspheres for diclofenac sodium dispersion via solvent evaporation. Ulcerogenic potential and low absorption restrict oral use of non-steroidal anti-inflammatory medicines such as diclofenac sodium [74]. Meltem and colleagues developed and tested Eudragit L100, poly(lactic acid-co-glycolic acid), and diclofenac sodium-containing nanoparticles. Diclofenac may cause severe side effects. Finally, Eudragit polymer nanoparticles sustained diclofenac sodium release [75]. Deepti Jain et al. employed Eudragit S100 microspheres to deliver peptide drugs such as insulin orally. A water-in-oil-in-water emulsion solvent evaporation technique was utilized to produce microspheres using polysorbate 20 as a dispersant and PVA/PVP as a stabilizer. In the manufacturing process, smaller interior aqueous-phase quantities (50 mL) and outer aqueous-phase volumes (25 mL) comprising PVA maximized encapsulation efficiency (81.8 ± 0.9%). At pH 1.0, maximally drug-encapsulated PVA-stabilized microspheres released 2.5 percent of insulin in two hours. In phosphate buffer (pH 7.4), microspheres released 22% in 1 h and 28% in 5 h. Interior and external aqueous phase volumes decrease with decreasing burst release. Higuchi kinetics release of drugs from microspheres. Figure 12 shows scanning electron microscopy of PVA-stabilized microspheres with smooth surfaces and a mean particle size of 32.51 ± 20 μm measured by laser diffractometry.

Oral PVA stabilized microspheres (6.6 IU insulin/kg animal weight) decreased blood glucose by 24% in normal albino rabbits, with a peak plasma glucose decrease of 76 ± 3.0% in 2 h and an effect lasting up to six hours. The percentage glucose reduction–time curve covers 93.75%. Thus, oral Eudragit S100 microspheres can protect insulin from proteolytic degradation in the gut and lower blood sugar. Eudragit S100 microspheres may be utilized to deliver peptide drugs such as insulin orally. Colon Eudragit S100 (insulin-loaded) microspheres decrease insulin secretion at low pH and delay insulin release at pH 7.4. In vivo microsphere studies showed that the polymer may protect insulin from proteolytic breakdown in the gastrointestinal system, resulting in hypoglycemia. Thus, Eudragit S100 microspheres are oral carriers for peptide medications such as insulin [76].

## 8. Applications of Eudragit in Biosensor

In pharmaceutical, clinical, and industrial fields, biosensors provide a simpler and cheaper alternative to complex bioanalytical systems. It is a self-contained medical device that tracks biological and chemical interactions at the transducer surface and gives quantitative and semi-quantitative analytical data. Enzymes, microorganisms, receptors, antibodies, and cells are biological detection components. Immobilization of bioreceptors and bio-elements on the electrode is essential to biosensor development. Nanomaterials made from Eudragit and other polymeric species and carbon species have unique physical and chemical properties such as covalent immobilization, cross-linking, physical adsorption, etc. The biosensor’s stability, conductivity, and sensitivity are improved using Eudragit nanomaterial [77]. In the last few years, there has been a lot of research going on to combine the unique properties of nanostructured materials with the chemical and physical properties of Eudragit. With this combination, it might be possible to make a new bioelectronic device that uses enzymes. When nanotechnology and polymeric materials are used together, very sensitive and quick assays can be made [78]. Ambra Giannetti and her colleagues showed that the Eudragit-coated Whispering Gallery Mode Resonator (WGMR) could be used as immunosensor. Following the demonstration, it was known to be the best tool for making optical biosensors. They have developed the most effective chemical protocol for making a thin layer of polymers that are all the same. Researchers in this study used Eudragit L100 coated micro spherical WGMR, which has been found to be a better choice. The goal of this experiment is to keep the resonator’s Q factor as high as possible. Fluorescence microscopy was used to figure out what was on with this change. It has been seen that the value of the Q factor stays the same when the globular protein IgG is chemically activated and covalently bound. This experiment also shows that the WGMR sensor can be used to find out how fast the analyte is binding [79]. Mamas Prodromidis and his team worked on a portable medical diagnostic device that uses a film-based biosensor made of responsive polymer (Eudragit S100) and a transducer for use at the point of care. They called it “BioPoC.” For this experiment to work, researchers used Eudragit S100 pH-responsive polymer because it is repeatable, inexpensive, and stable. They also made a device out of very cheap materials to keep the cost down. The BioPoC device is very selective when it comes to elements of the matrix. This device is very accurate when it comes to qualitative diagnostic tests. For example, it can find H. Pylori in a gastric antrum biopsy conducted close to the patient and measured the amount of urea in a person’s undiluted urine. Mamas Prodromidis and his colleagues came to the conclusion that making biosensors based on how polymers break down has more benefits in terms of productivity, the accuracy of results, and ease of use [80]. In the article “Direct electrochemical biosensing in gastrointestinal fluids,” the pH-sensitive polymers Eudragit L100 and Eudragit E PO are used in biomedical devices. Researchers used edible electrochemical biosensors made from things such as olive oil and activated charcoal to protect the activity of the glucose oxidase (GO) enzyme from strongly acidic conditions. The enzyme’s resistance to being deactivated by a low pH allows for direct glucose monitoring in a strong acidic area for up to 90 min, while the response of a traditional screen-printed biosensor drops significantly after 10 min in the same fluid. In the edible biosensors that were made, the electrode surface was coated with Eudragit L100 and E PO grades that respond to pH. These polymers make it possible for sensors to be activated by fluid in the intestines and stomach at a certain time. It showed a linear response to glucose between 2 and 10 mM, with sensitivity depending on the pH of the GI fluid. The combination of an enteric-coated enzyme system and an enzyme system that is resistant to pH gives stability and protection to the biosensor. This improves treatments such as pancreatitis that depend on getting enzymes into the GI tract. The possible mechanical use of the research findings can be utilized to make an ingestible capsule for real-time monitoring of analyte in the GI tract [81].

## 9. Patent on Eudragit-Based Pharmaceutical Formulation for Drug Delivery

Researchers are now working on nanotechnology compositions to enhance pharmaceutical therapeutic potential. Eudragit-based compositions get several patents annually. Due to its biological and physicochemical features, Eudragit is continuously being used. We created a Eudragit-based formulation table using previously reported pharmaceutical patents in our article. We examined Google patents and associated studies. Table 4 summarizes recent patents.

## 10. Future Prospective

In the first generation of drug delivery agents, Eudragit has gained a notable name as an effective coating for tablets, a flavoring agent, and an enteric coating material for drugs that were intended to be absorbed by the gastrointestinal system. In a similar fashion, the second generation of drug delivery investigated the use of Eudragit for coating microspheres and nanocarriers in order to provide pH-sensitive chemotherapeutic targeting. The primary goal of these two generations of drug delivery systems is to overcome obstacles caused by formulation and the physicochemical characteristics of the medication. On the other hand, the fundamental purpose of the third generation of drug delivery employing Eudragit is to overcome biological boundaries. Eudragit was an important part of the process that led to the creation of a variety of dosage forms and the enhancement of the delivery qualities of a large number of active pharmacological agents. In the beginning, it was employed with tablet excipients and tablet coating, but later on, it was processed into nanocarriers for the purpose of targeting the acidic tumor microenvironment. In the not-too-distant future, Eudragit will find uses as a material for 3D printing, nanofibers for dressing, tissue engineering, and other biomedical processes. In order to synthesize various grades of Eudragit, an innovative technique that involves the improvement of the polymerization process and catalysis for the purpose of changing physicochemical qualities may be used. It is possible to employ the formulation of unique, beneficial, and helpful Eudragit combinations for the formulation of dosage forms with gold standard zero order release, which is challenging in the field of drug delivery.

## 11. Conclusions

In this overview, we have summarized various Eudragit polymer grades available in the literature. Eudragit is a synthetic polymethacrylate copolymer with a wide range of hydrophobicity and hydrophilicity, as well as a variety of functional group tolerance and the ability to cross link in a number of different environments. Because of this, Eudragit polymer grades are the gold standard for treating a wide range of illnesses with medication. The prolonged release of Eudragits, which are synthetic polymers that are harmless, may be tailored to individual needs by combining different grades. Researchers may get a foundational knowledge of the many formulations possible using Eudragit from the above-listed articles, which explains a wide range of medications and processes involved in these preparations. Despite the many published experimental findings, there is still a great deal of independent study to be conducted to uncover its vast and growing range of uses and pharmacological therapeutic advances.

## Data Availability

This research did not report any data.

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
