# Peer review of "A Systematic Overview of Eudragit® Based Copolymer for Smart Healthcare"

_pharmaceutics, 2023, doi:10.3390/pharmaceutics15020587_

Round 1

Reviewer 1 Report (Previous Reviewer 1)

In the revised manuscript, the authors summarize Eudragit® from molecule to material design and its applications, which enrich the drug delivery systems for precise release and targeting. The authors have revised the manuscript according to the comments. Therefore, I would recommend this manuscript for publication. However, there are still some questions.

1.     Please clarify the scope of “Gene Based Drug Delivery”, why the part “6.2. Cancer Based Drug Delivery” is included? In the manuscript, the way that use Eudragit based drug delivery to treat cancer is not all based on genes.

2.     As the author mentioned, “Gene delivery system is most important application for successful delivery of the siRNA, DNA, RNA, Plasmid and small genes”, protein/peptide should not be included in gene delivery system. It is suggested to correct the subtitle more accurately.    

3.     Some minor format problems should be checked carefully, i.e., the reference 12 (Line 822), the abbreviation of PLGA (Line 780).

Author Response

(Reviewer 1)

  1. Please clarify the scope of “Gene Based Drug Delivery”, why the part “6.2. Cancer Based Drug Delivery” is included? In the manuscript, the way that use Eudragit based drug delivery to treat cancer is not all based on genes.

Response: Thank you for valuable comments. We corrected all subheading from Gene based drug delivery to easy follow up for reader.

  1. As the author mentioned, “Gene delivery system is most important application for successful delivery of the siRNA, DNA, RNA, Plasmid and small genes”, protein/peptide should not be included in gene delivery system. It is suggested to correct the subtitle more accurately.

Response: Protein/peptide moved to other subsection in gene delivery system.

  1. Some minor format problems should be c
  2.  
  3. hecked carefully, i.e., the reference 12 (Line 822), the abbreviation of PLGA (Line 780).

Response: We have made changes in line 822 and 780 as per reviewer suggestion.

Reviewer 2 Report (Previous Reviewer 2)

I reviewed a previous version of this article. Comparing them, I can say that the manuscript has been significantly revised and improved.

The answer to my comments satisfied me.

The only thing I still don't really like is the style in which the schemes are drawn. Indexes denoting the number of atoms are sometimes in strange places.

However, my opinion is that this article should be accepted for publication.

Author Response

I reviewed a previous version of this article. Comparing them, I can say that the manuscript has been significantly revised and improved.

The answer to my comments satisfied me.

The only thing I still don't really like is the style in which the schemes are drawn. Indexes denoting the number of atoms is sometimes in strange places.

However, my opinion is that this article should be accepted for publication.

Response: We have made changes in all schemes style and indexes denoting the number of atoms as per reviewer suggestion.

Reviewer 3 Report (Previous Reviewer 3)

Authors have significantly revised the manuscript and addressed comments. It could be accepted now. 

Author Response

Authors have significantly revised the manuscript and addressed comments. It could be accepted now. 

Reviewer 4 Report (Previous Reviewer 4)

The manuscript entitled: A Systematic Overview of Eudragit® Based Copolymer for Smart Healthcare of the authors: Aniket Nikam, Priya Ranjan Sahoo, Roshani Pagar, Shubham Muasale, Ana Cláudia Paiva-Santos, and Prabhanjan S. Giram was improved.

I believe that the paper may be suitable for publication in the MDPI journal Pharmaceutics after addressing the following considerations listed below.

- In line 30, I suggest that “Synthesis” to be written in lowercase letters.

- In lines 72 and 73, I suggest that “Poly (butyl methacrylate-co-(2-dimethylamino) ethyl methacrylate-co-methyl methacrylate)” to be written “Poly(butyl methacrylate-co-(2-dimethylamino)ethyl methacrylate-co-methyl methacrylate)” (without space between poly and parenthesis and between parenthesis and ethyl).

- Names of all polymers or copolymers should be written without spaces between poly and parentheses.

- In line 89, I suggest that Low” to be written in the lowercase letter.

- I suggest that all headings be written using the title case, e.g in line 232, I suggest be written: ”4.1. Atom Transfer Radical Polymerization” instead of “4.1. Atom transfer radical Polymerization”.

In Table 1, I suggest to be written:

- “Cationic (Aminoalkyl methacrylate copolymers)” instead of “Cationic (Aminoalkylmethacrylate copolymers)”

- each Application without a point, e.g. “Increased geriatric and pediatric patient compliance” instead of “Increased geriatric and pediatric patient compliance.

- in the "Chemical Composition" column, the corrected names of the copolymers, for example:

“Poly(butyl methacrylate-co-2-(dimethylamino)ethyl methacrylate-co-methyl methacrylate) 1:2:1” instead of “Poly (butyl methacrylate, (2-di-methyl aminoethyl) methacrylate, methyl methacrylate) 1:2:1”

“Poly(methacrylic acid-co-methyl methacrylate) 1:1” instead of “Poly (methacrylic acid, methyl methacrylate) 1:1”.

“Poly(ethyl acrylate-co-methyl methacrylate-co-2-trimethylammonioethyl methacrylate chloride) 1:2:0.2” instead of “Poly (ethyl acrylate, methyl methacrylate, 2-trime-thylammonioethyl methacrylate chloride” or “2-(methacrylo-yloxy)-N,N,N-trimethylethanaminium chloride) 1:2:0.2 ”

- “Anionic (Methacrylic acid copolymers)” instead of “Anionic (Methacrylic copolymers)”

- “Neutral (Ammonioalkyl methacrylate copolymers)” instead of “Neutral (Ammonioalkyl methyacrlate copolymers)”

- “Neutral (Alkyl methacrylate copolymer)” instead of “Neutral (Methacrylic acid copolymers)”.

- In Scheme 1, it is necessary that in the copolymer structures, the number of atoms or groups of the same type is written as a subscript, for example, the number of H atoms in the NH2. In the case of H3C group, the number of H atoms it is necessary to be written on the right side of the H atom (instead of the left side of H, 3HC). Likewise, throughout the document. In structure of 2,2'-disulfanediyldinicotinic acid, it is necessary that the COOH group to be written in one row (instead of two rows).

- In line 160, it is necessary that “thiolated poly (methacrylic acid-co-ethyl acrylate” to be replaced with “thiolated poly(methacrylic acid-co-ethyl acrylate)”.

- In Scheme 2, it is necessary that:

- in the case of CH2 group, the number of H atoms to be written on the right side of the H atom (instead of above the H atom)

-“N2” to be written “N2”.

- In line 177, it is necessary that “N,N-dicyclohexylcarbodiimide” to be replaced with “N,N-dicyclohexylcarbodiimide”.

- In Scheme 3, it is necessary that:

-“NH2” to be written “NH2

-“2HN” to be written “H2N”

-“Et3N” to be written “Et3N”.

- In lines 185 and 186, it is necessary that “2-methoxy-N4-phenyl-1,4-phenylenediamine” to be replaced with “2-methoxy-N4-phenyl-1,4-phenylenediamine”.

- In Scheme 5, it is necessary that:

-“NH2” to be written “NH2

-“2HN” to be written “H2N”

- In Scheme 6, it is necessary that:

- in the case of CH2 group, the number of H atoms to be written on the right side of the H atom (instead of above the H atom)

-“2HC” to be written “H2C”.

- In line 279, it is necessary that poly(-caprolactone)“ to be replaced with “poly(ε-caprolactone)”.

- In line 286, it is necessary that poly (ε-caprolactone“ to be replaced with “poly(ε-caprolactone)”.

- In line 287, it is necessary that poly(ε-caprolactone“ to be replaced with “poly(ε-caprolactone)”.

- In line 292, it is necessary that hydroxymethylpropylcellulose” to be replaced with “hydroxypropylmethylcellulose”.

- In lines 625 and 626, I suggest that “Eudragit L 100 poly (lactic acid-co-glycolic acid” to be written “Eudragit L 100, poly(lactic acid-co-glycolic acid)”.

In the “Abbreviations” section, please writte:

- “poly(lactic acid-co-glycolic acid)” instead of “poly (lactic acid-co-glycolic acid”

- “5-fluorouracilinstead of “5-fluourouacil”. The same correction is needed in Table 3.

Author Response

I believe that the paper may be suitable for publication in the MDPI journal Pharmaceutics after addressing the following considerations listed below.

- In line 30, I suggest that “Synthesis” to be written in lowercase letters.

Response: We have made changes in line 30.

- In lines 72 and 73, I suggest that “Poly (butyl methacrylate-co-(2-dimethylamino) ethyl methacrylate-co-methyl methacrylate)” to be written “Poly(butyl methacrylate-co-(2-dimethylamino)ethyl methacrylate-co-methyl methacrylate)” (without space between poly and parenthesis and between parenthesis and ethyl).

Response: We have made changes in lines 72 and 73.

- Names of all polymers or copolymers should be written without spaces between poly and parentheses.

Response: We have made changes in all polymers or copolymers written without spaces between poly and parentheses.

- In line 89, I suggest that “Low” to be written in the lowercase letter.

Response: We have made changes in line 89.

- I suggest that all headings be written using the title case, e.g in line 232, I suggest be written: ”4.1. Atom Transfer Radical Polymerization” instead of “4.1. Atom transfer radical Polymerization”.

Response: We have made changes in line 232.

In Table 1, I suggest to be written:

- “Cationic (Aminoalkyl methacrylate copolymers)” instead of “Cationic (Aminoalkylmethacrylate copolymers)”

Response: We have made changes in manuscript as per reviewer suggestion.

- each Application without a point, e.g. “Increased geriatric and pediatric patient compliance” instead of “Increased geriatric and pediatric patient compliance.”

Response: We have made changes in manuscript as per reviewer suggestion.

- in the "Chemical Composition" column, the corrected names of the copolymers, for example:

“Poly(butyl methacrylate-co-2-(dimethylamino)ethyl methacrylate-co-methyl methacrylate) 1:2:1” instead of “Poly (butyl methacrylate, (2-di-methyl aminoethyl) methacrylate, methyl methacrylate) 1:2:1”

Response: We have made changes in manuscript as per reviewer suggestion.

“Poly(methacrylic acid-co-methyl methacrylate) 1:1” instead of “Poly (methacrylic acid, methyl methacrylate) 1:1”.

Response: We have made changes in manuscript as per reviewer suggestion.

“Poly(ethyl acrylate-co-methyl methacrylate-co-2-trimethylammonioethyl methacrylate chloride) 1:2:0.2” instead of “Poly (ethyl acrylate, methyl methacrylate, 2-trime-thylammonioethyl methacrylate chloride” or “2-(methacrylo-yloxy)-N,N,N-trimethylethanaminium chloride) 1:2:0.2 ”

Response: We have made changes in manuscript as per reviewer suggestion.

- “Anionic (Methacrylic acid copolymers)” instead of “Anionic (Methacrylic copolymers)”

Response: We have made changes in manuscript as per reviewer suggestion.

- “Neutral (Ammonioalkyl methacrylate copolymers)” instead of “Neutral (Ammonioalkyl methyacrlate copolymers)”

Response: We have made changes in manuscript as per reviewer suggestion.

- “Neutral (Alkyl methacrylate copolymer)” instead of “Neutral (Methacrylic acid copolymers)”.

Response: We have made changes in manuscript as per reviewer suggestion.

- In Scheme 1, it is necessary that in the copolymer structures, the number of atoms or groups of the same type is written as a subscript, for example, the number of H atoms in the NH2. In the case of H3C group, the number of H atoms it is necessary to be written on the right side of the H atom (instead of the left side of H, 3HC). Likewise, throughout the document. In structure of 2,2'-disulfanediyldinicotinic acid, it is necessary that the COOH group to be written in one row (instead of two rows).

Response: We have made changes in scheme 1 as per reviewer suggestion. 

- In line 160, it is necessary that “thiolated poly (methacrylic acid-co-ethyl acrylate” to be replaced with “thiolated poly(methacrylic acid-co-ethyl acrylate)”.

Response: We have made changes in line 160.

- In Scheme 2, it is necessary that:

- in the case of CH2 group, the number of H atoms to be written on the right side of the H atom (instead of above the H atom)

Response: We have made changes in scheme 2 CH2 group, the number of H atoms to be written on the right side of the H atom

-“N2” to be written “N2”.

Response: We have made changes in scheme 2 we have written N2 instated of N2. 

- In line 177, it is necessary that “N,N-dicyclohexylcarbodiimide” to be replaced with “N,N′-dicyclohexylcarbodiimide”.

Response: We have made changes in line 177.

- In Scheme 3, it is necessary that:

-“NH2” to be written “NH2”

-“2HN” to be written “H2N”

-“Et3N” to be written “Et3N”.

Response: We have made changes in scheme 3 we have written NH2 instated of NH2, 2HN instead of H2N as well as Et3N is instead of Et3N.

- In lines 185 and 186, it is necessary that “2-methoxy-N−4-phenyl-1,4-phenylenediamine” to be replaced with “2-methoxy-N4-phenyl-1,4-phenylenediamine”.

Response: We have made changes in line 185 and 186.

- In Scheme 5, it is necessary that:

-“NH2” to be written “NH2”

-“2HN” to be written “H2N”

Response: We have made changes in scheme 5 we have written NH2 instated of NH2, 2HN instead of H2N.

- In Scheme 6, it is necessary that:

- in the case of CH2 group, the number of H atoms to be written on the right side of the H atom (instead of above the H atom)

-“2HC” to be written “H2C”.

Response: We have made changes in scheme 6 we have written 2HC instead H2C.

- In line 279, it is necessary that “poly(-caprolactone)“ to be replaced with “poly(ε-caprolactone)”.

Response: We have made changes in line 279.

- In line 286, it is necessary that “poly (ε-caprolactone“ to be replaced with “poly(ε-caprolactone)”.

Response: We have made changes in line 286.

- In line 287, it is necessary that “poly(ε-caprolactone“ to be replaced with “poly(ε-caprolactone)”.

Response: We have made changes in line 287.

- In line 292, it is necessary that “hydroxymethylpropylcellulose” to be replaced with “hydroxypropylmethylcellulose”.

Response: We have made changes in line 292.

- In lines 625 and 626, I suggest that “Eudragit L 100 poly (lactic acid-co-glycolic acid” to be written “Eudragit L 100, poly(lactic acid-co-glycolic acid)”.

Response: We have made changes in line 625 and 626.

In the “Abbreviations” section, please writte:

- “poly(lactic acid-co-glycolic acid)” instead of “poly (lactic acid-co-glycolic acid”

- “5-fluorouracil” instead of “5-fluourouacil”. The same correction is needed in Table3.

Response: We have made changes in abbreviations section and Table 3.

This manuscript is a resubmission of an earlier submission. The following is a list of the peer review reports and author responses from that submission.

Round 1

Reviewer 1 Report

It is a hot topic in the research of drug delivery systems that manufacturing of drugs for precise release and targeting. The authors reviewed the Eudragit® from molecule to material design and its applications. However, in the application part, the author's description is not clear, and logical organization and classification need to be improved (i.e., why section “5.2.2” was included in "5.2. GENE BASED DRUG DILIVERY"? As the authors mentioned in part 5.2 in the manuscript, " successful delivery of the siRNA, DNA, RNA, Plasmid and small genes", protein/peptide exceeds the above scope). Furthermore, there is no subtitle “5”. In addition, some sections need to be supplemented with more references, such as 5.1.2, the reference [25] needs to be more refined. In a whole, the manuscript cannot be accepted.

Other comments:

1. It is recommended to mark the scheme and figure in the main text.

2. Subtitle should be clear (the section 4 and 5).

3. It is recommended to explain more in detail about “The limitations due to the limitation observed during drugs transport through the gastrointestinal mucosa. (Line 148)” What is the limitation?

4. Based on what principle that mucoadhesive nanoparticles helped temporarily open tight junctions between epithelial cells (Line 449). It is better to give the reference.

5. There are two “Table 2” (Line 132, Line 455).

Reviewer 2 Report

This review combines material on a rather interesting and relevant topic. 

Today, the design developed by the German concern Evonik Pharma GmbH and registered under the subname EUDRACOL® is the first commercial product on the market that uses the principle interpolymer interaction of oppositely charged Eudragit® copolymers for targeted drug delivery.

This review is well written and has an impressive list of references. However, I have a number of questions and comments:

1. I know that Ruslan I. Moustafin is doing extensive research on this topic. However, only one reference is given to it in the bibliography.

2. Table 3 (line 455) is erroneously called Table 2 and is not referenced in the text.

3. Table 4 (line 700) is erroneously called Table 3.

4. On the Schemes throughout the text of the manuscript, the numbers indicating the number of atoms of elements should be indicated in lower case (for example, CH3 instead of CH3).

5. The list of references is not very detailed. In some paragraphs, the name of the journal, DOI and the names of the authors are missing, and in some, even the title is not completed to the end. Review links numbered 10, 12, 13, 18, 19, 20, 21, 22, 24, 28, 29, 30, 31, 34, 37-41, 43,  45-48, 58, 61, 62, 66.

6. Headings and subheadings throughout the text have different formatting. It is required to make them in the same style.

7. English needs improvement.

Some of my suggested corrections are in the attached file.

The comments I have made do not affect the significance of the work. I recommend it for publication after revision.

Reviewer 3 Report

Giram and co-workers summarizes the developments of Eudragit® based copolymers for smart healthcare. The manuscript is interesting and could be considered for publication after major revisions.

1. Abstract is too brief. Please elaborate.

2. Drawing quality is extremely poor and not consistent.

3. The introduction is one paragraph. Please split it into multiple for easy understanding.

4. Presentation of references is not according to format such as [6][7], revise to [6,7] and check the whole manuscript.

5. The resolution of figures is extremely poor.

6. Multiple corrections required. e.g. subscript and superscript, use of capital letters in the middle of sentences etc.

7. Heading style is inconsistent. 

Reviewer 4 Report

In this manuscript entitled: A Systematic Overview of Eudragit® Based Copolymer for Smart Healthcare, the authors: Aniket Nikam, Priya Ranjan Sahoo, Roshani Pagar, Shubham Muasale, Ana Cláudia Paiva-Santos, and Prabhanjan S. Giram reviewed the literature on Eudragit®, from molecule to material design, its characterization, and its applications in healthcare.

I believe that the paper may be suitable for publication in the MDPI journal Pharmaceutics after addressing the following considerations listed below.

- Keywords, except for Eudragit, I suggest to be written in lowercase letters.

- In lines 57-59, I suggest that the words: “Reversible”, “Atom” and “Groupto be written in lowercase letters.

- In line 64, I suggest that methacrylates” to be written “alkyl methacrylates”.

- In line 64, “dimethyl aminoethyl” is “2-(dimethylamino)ethyl methacrylate”? In this case, it is necessary to correct it.

- In line 71, I suggest that “dimethyl aminoethyl methacrylate” to be written “dimethylaminoethyl methacrylate”.

- In lines 71 and 72, I suggest that “Poly (butyl methacrylate-co-(2-dimethylaminoethyl) methacrylate-methyl methacrylate” to be written “Poly(butyl methacrylate-co-(2-dimethylamino)ethyl methacrylate-co-methyl methacrylate)”.

- In all manuscript, it is necessary to be a space between the value of temperature and “°C”, for example in lines 76, 79, and 82.

- In line 85, it is necessary to be corrected: “1500C”.

- In line 88, I suggest that Low” to be written in the lowercase letter.

- In line 89, it is necessary that “400C and 550C” to be written “40 °C and 55 °C”.

- In line 90, it is necessary to be corrected: “700C”.

- I suggest that all headings be written using the title case, e.g. in line 98, I suggest be written: ”2. Classification of Eudragit Polymer” instead of “2. CLASSIFICATION OF EUDRAGIT POLYMER”, in line 219, I suggest be written: ”4.1. Atom Transfer Radical Polymerization” instead of “4.1. Atom transfer radical Polymerization”.

- In line 102, I suggest that Table 1” to be written without bold. Likewise, throughout the document.

In Table 1, I suggest to be written:

- “Aminoalkyl methacrylate copolymers” instead of “Aminoalkylmethacrylate copolymers”

- first column: “EUDRAGIT Grade”

- each Application without a point, e.g. “Increased geriatric and pediatric patient compliance” instead of “Increased geriatric and padiatric patient compliance.”

- in the "Chemical Composition" column, first the corrected copolymer name, then the monomer ratio, for example:

“Poly(butyl methacrylate-co-2-(dimethylamino)ethyl methacrylate-co-methyl methacrylate) 1:2:1” instead of “Poly 1:2:1(butyl methacrylate, (2-dimethyl aminoethyl) methacry-late, methyl methacrylate)”

“Poly(methacrylic acid-co-methyl methacrylate) 1:1” instead of “Poly 1:1(methacrylic acid, methyl methacrylate)”.

“Poly(ethyl acrylate-co-methyl methacrylate-co-2-trimethylammonioethyl methacrylate chloride) 1:2:0.2” instead of “Poly1:2:0.2(ethyl acrylate, methyl methacrylate, trimethyl aminoethyl methacrylate chloride)”

- “Methacrylic acid copolymers” instead of “Methacrylic copolymers”

- “Ammonioalkyl methacrylate copolymers” instead of “Ammonioalkyl methyacrlate copolymers”

- “Alkyl methacrylate copolymer” instead of “Methacrylate copolymer”.

- I need that “trimethyl aminoethyl methacrylate chloride” to be replaced with “2-trimethylammonioethyl methacrylate chloride” or “2-(methacryloyloxy)-N,N,N-trimethylethanaminium chloride”.

- Throughout the document, it is necessary to merge the citations, e.g., in line 119, I suggest to be written “[4,5]” instead of “[4][5]”.

- In line 120, it is necessary that “shown” to be written instead of  “showed” and a point after “Figure 1”.

- In line 122, I suggest that to be a point after “Figure 2” and “here” to be written with a capital letter.

- In line 125, I suggest that “demonstrated” to be replaced with “are presented” and “the spectral range” to be replaced with “The spectral ranges”.

- In line 147, it is necessary that “route. there” to be replaced with “route, there”.

- In Scheme 1, it is necessary that in the copolymer structures, the number of atoms or groups of the same type is written as a subscript, for example, the number of H atoms in the CH3 group and the number of ethyl groups in Et3N. Likewise, throughout the document.

- In line 154, it is necessary that “thiolated poly (methacrylic acid-co-ethyl acrylate” to be replaced with “thiolated poly(methacrylic acid-co-ethyl acrylate)”.

- In line 171, it is necessary that “N, N′-dicyclohexyl carbodiimide and N-hydroxyl succinimide” to be replaced with “N,N′-dicyclohexylcarbodiimide and N-hydroxysuccinimide”.

- In lines 179 and 180, it is necessary that “2-methoxy-N4-phenyl-1,4-phenylenediamine” to be replaced with “2-methoxy-N4-phenyl-1,4-phenylenediamine”.

- In lines 180 and 181, it is necessary that “1-Ethyl-3-(3-dimethylaminopropyl) carbodiimide (EDC) and 1,1′-Carbonyldiimidazole (CDI)” to be replaced with “1-ethyl-3-(3-dimethylaminopropyl)carbodiimide (EDC) and 1,1′-carbonyldiimidazole (CDI)”.

- In line 206, is no need for a space between “N,” and “ N′ “.

- In line 210, a point is missing.

- In Scheme 6, it is necessary that the structure of the acrylated EPO to be corrected. The N atom to which the acryloyl group is linked should have a positive charge and the Cl atom should have a negative charge.

- In line 230, I suggest that “kp” to be written “kp”.

- In line 263, I suggest that “Ophthalmic” to be written in the lowercase letter.

- In line 266, I suggest that “Hydrogels” to be written in the lowercase letter.

- In lines 268, 275, and 276, it is necessary that poly(-caprolactone)“ to be replaced with “poly(ε-caprolactone)”.

- In line 281, “Hydroxymethyl Propyl Cellulose” is “hydroxypropylmethylcellulose”? In this case, it is necessary to correct it.

- In line 287, I suggest that “Losartan potassium” to be written “losartan potassium”. In line 292, etc., this suggestion applies to “Tenofovir”.

- In line 292, it is necessary that “For” to be written in the lowercase letter.

- In line 295, I suggest that (SEM)” to be written without parentheses.

- In lines 299 and 300, I suggest that hours” to be abbreviated “h”.

- In line 326, I suggest that Aung NN et al.” to be written “Aung et al.”.

- In lines 328, 334, and 344, it is necessary that “polyvinyl pyrrolidone” to be written “polyvinylpyrrolidone”.

- In lines 357 and 358, I suggest that the first sentence to be reformulated: “A diagram of how the pH of the wound influences the release mechanism is shown in Figure 5.”

- In line 369, I suggest that Swellable” to be written in the lowercase letter.

- In line 382, a space is necessary between “[29].” and “Nanofibers”.

- In line 487, it is necessary that “Poly (lactide-co-glycolide; PLGA)” to be written “poly(lactide-co-glycolide), PLGA,”.

- In line 566, it is necessary that “Invitro” to be written: “In vitro”.

- In line 582, it is necessary that “for 5-Fluorouracil colon targeting (5-FU)” to be replaced with “for 5-fluorouracil (5-FU) colon targeting” or “for targeting the colon with 5-fluorouracil (5-FU)”.

- In line 591, it is necessary that ”target 5-Fluorouracil in the colon (5-FU)” to be replaced with ”target 5-fluorouracil (5-FU) in the colon”.

In line 614, I suggest that “poly (lactic-co-glycolic acid)” to be written “poly(lactic acid-co-glycolic acid)”.

- In line 641, I suggest that the heading be written ”5.4. Application of Eudragit in Biosensor” instead of “5.4. APPLICATION OF EUDRAGIT IN BIOSENSOR:”.

The same observations are available for the “ABBREVIATIONS” section.